# Practical Review on Preclinical Human *3D* Glioblastoma Models: Advances and Challenges for Clinical Translation

**DOI:** 10.3390/cancers12092347

**Published:** 2020-08-19

**Authors:** Aurélie Soubéran, Aurélie Tchoghandjian

**Affiliations:** CNRS, INP, Institute of Neurophysiopathology, Aix-Marseille University, 13005 Marseille, France; aureliesouberan@gmail.com

**Keywords:** glioblastomas, preclinical *3D* cancer models, spheroids, tumorospheres, organotypic slices, explants, tumoroids, organoids

## Abstract

Fifteen years after the establishment of the Stupp protocol as the standard of care to treat glioblastomas, no major clinical advances have been achieved and increasing patient’s overall survival remains a challenge. Nevertheless, crucial molecular and cellular findings revealed the intra-tumoral and inter-tumoral complexities of these incurable brain tumors, and the essential role played by cells of the microenvironment in the lack of treatment efficacy. Taking this knowledge into account, fulfilling gaps between preclinical models and clinical samples is necessary to improve the successful rate of clinical trials. Since the beginning of the characterization of brain tumors initiated by Bailey and Cushing in the 1920s, several glioblastoma models have been developed and improved. In this review, we focused on the most widely used *3D* human glioblastoma models, including spheroids, tumorospheres, organotypic slices, explants, tumoroids and glioblastoma-derived from cerebral organoids. We discuss their history, development and especially their usefulness.

## 1. Introduction

Glioblastomas (GBM) are lethal brain tumors. They are classified as grade IV gliomas by the World Health Organization as they are highly proliferative, infiltrative, necrotic, neoangiogenic and are associated with an immunosuppressive environment [1]. The therapeutic options for GBM are limited and mainly consist in a surgery followed by adjuvant radiotherapy and chemotherapy based on the alkylating agent temozolomide [2]. This standard of care treatment has remained unchanged since 2005, even though thousands of clinical trials were performed since then. Treating GBM remains a therapeutic challenge although great advances have been achieved concerning the knowledge of this cancer. The molecular and cellular heterogeneity of GBM has been well characterized and single cell analyses revealed inter- and intra-tumoral heterogeneity underlying the complexity of treating these tumors. Tumors are an ecosystem in which tumor cells and cells of the microenvironment are closely interacting, creating a bidirectional dependency. The GBM microenvironment consists of microglia, bone marrow derived macrophages, astrocytes, oligodendrocytes, neurons, glial and neuronal progenitors, pericytes, endothelial cells and extracellular matrix [3]. This complex and heterogeneous entity forms a network which produces a microenvironment favorable to tumor growth, invasion, resistance to therapeutics and immune escape [4,5]. Therefore, to achieve therapeutic success, cancer cells—including cancer stem cells—must be eliminated, but the tumor microenvironment has also to be considered as a main target.

Clinical developments are based on results generated by experimental protocols used in preclinical studies. Appropriate models should reproduce the GBM characteristics studied, be adapted for quantification of data, while considering cellular and molecular heterogeneities, differences in accessibility to nutrients as well as O_2_ and pH variations if necessary. However, the scientific question behind the study determines the choice of the model to be used, whose accuracy and proper setting up is crucial to draw the right conclusions.

To mimic the GBM composition, organization, physical constraints, drug resistance/sensitivity and drug penetration, several *3D* human-derived GBM models exist. The first *3D* human GBM model was established in 1929 and consisted of small pieces of tumors, named explants, derived from human GBM [6]. The following 3D model was then developed in 1983 with the generation of the first GBM spheroids [7]. Since then, a succession of models have been proposed: tumoroids, which are generated from a piece of GBM tissue or from cancer stem-like cells (CSLCs); organotypic slices, which consists in grafting spheroids or isolated cells on brain slices; the tumorospheres, based on the discovery of CSLCs; and GBM-derived from cerebral organoids (Figure 1).

Some of these models are simple and made of cancer cells only while others are more complex, composed of several cell types while displaying characteristics of the parental tumor. Confusion may arise because the terminology used to name each model often differs from one article to another (Table 1). To help choosing the more accurate tool to address the raised question as well as understanding suitable experiments that can be performed with each model, we describe in this review the culture conditions and the uses of the main *3D* human GBM models described in the literature. The models were classified according to their initial biological material and their purpose, and will be organized into the following three groups: the sphere-based models derived only from established or primary tumor cell lines, the organotypic cultures based on tissue culture, and organoids generated to reconstitute a tumor in a dish.

## 2. Sphere-Based Models

### 2.1. Spheroids

Spheroids are the most widely used model. They consist in the culture of established GBM cell lines in which the cells grow as spheres either in suspension or in a special matrix in appropriate culture medium.

The practice of culturing cells as aggregates was initially developed by Sutherland and collaborators in 1971 to study the sensitivity of tumor cells to radiotherapy. They reported it as a realistic in vitro model of tumor growth displaying morphological, functional, and mass-transport properties of the corresponding tissue and therefore suitable for radiobiology studies [29].

Two main techniques (Table 2) are used to generate spheroids: the “non-scaffold” methods and the “scaffold” methods. The “non-scaffold” methods rely on the use of matrix-coated plates or hydrophobic polymer-coated plates to counteract cell attachment. Alternatively, spheroids can grow into a drop of medium or in a shaking device called “spinner flask bioreactor”. The “scaffold” methods use hydrogels, bioreactors, or other synthetic extra-cellular matrix-like structures to induce cell growth as a *3D* culture. The use of a scaffold might introduce a bias and influence spheroids growth and behavior. As an example, alginate membranes could reduce oxygen, nutrient supply, and cell-cell contacts. In comparison, oil-free hydrogels allow free flow of nutrients [30]. Moreover, matrixes made of collagen I promote cell invasion but decrease spheroids growth rate [31]. The most commonly used culture media for spheroids are Eagle’s minimal essential medium (EMEM), Dulbecco modified Eagle’s minimal essential medium (DMEM) and Roswell Park Memorial Institute medium (RPMI) supplemented with penicillin/streptomycin and fetal bovine serum (FBS). Human serum has also been used to better mimic the natural microenvironment of cancer cells [32,33]. The majority of the existing GBM cell lines (U-87MG, U-105MG, U-118MG, U-138MG, U-178MG, U-251MG, U-343MG, U-373MG, U-1231MG, BMG-1, T98G, A172, SNB-19, ACBT, M059J, LN-229, Hu197, and LN-18) are able to form spheroids except the U-343MG cell line [7,34,35,36,37,38,39,40,41,42,43,44,45]. Spheroids growth is standardized according to the number of cells plated and can be divided into the following three phases: the initial growth (days 1–3), the plateaued volume (days 3–4), and the rapid secondary growth (days 4–6) [46].

Spheroids are organized structures composed of several layers. The external layers are accessible to nutrients and oxygen and contain proliferative cells, the intermediate layers are composed of senescent cells, and the core of the spheroid is mainly necrotic [53]. Therefore, gradients of proliferation, oxygen, nutrients and pH can be observed from the external layer to the inner part of the spheroid [7,34,36]. Spheroids can produce an organized extracellular matrix, more abundant in the core than in the periphery, and composed by fibronectin, laminin, collagen and glycosaminoglycans. Extracellular matrix could be organized into both fibrillar and nonfibrillar structures [54].

The background acquired during the last 50 years in GBM research, makes this model the standard *3D* model to study tumor growth and therapies. Spheroids are appropriate tools for drug testing and high-throughput drug screening [36,50,55,56]. They have been used to investigate mechanisms involved in anticancer drug resistance [57,58] and GBM cell invasion.

Spheroids have the advantage to be easy to maintain and to be easy to use for genetic manipulation. Nevertheless, they have several limitations. They poorly represent the primary tumor biology and share little histological resemblance to it [59]. Discrepancies between genome and gene expression of spheroids and primary tissues have been observed such as the onset of new recurrent aberrations in spheroids [60]. Furthermore, the human-derived immortalized GBM cell lines used to generate spheroids (e.g., U-87MG and U-251MG) have to be used with care due to the lack of similarity to actual GBM tumor cells and because of their controversial origins [61].

### 2.2. Tumorospheres

The tumorosphere model relies on the capability of cells with stem-like cell properties to self-renew. Stem-like cells grow clonally as free-floating spheres in a defined medium. Tumorospheres are formed by the symmetric or asymmetric division of the stem-like cells to generate other stem-like cells or cells more engaged in a pathway of differentiation, respectively.

The tumorosphere model allows the identification of cells with stem-like cell properties in normal and tumoral brain tissues. It is a *3D* culture model based on the neurosphere model described by Reynolds and Weiss, which derived from non-neoplastic neural stem cells from the adult mouse brain [62]. Singh and collaborators were the first to use it in primary brain tumors (medulloblastoma, pilocytic astrocytoma, ependymoma, ganglioglioma) to identify CSLCs. In GBM, these CSLCs have been described to be enriched in the CD133 population [63], but later several other surface markers have been identified (A2B5 [11,64], L1CAM [65], integrin α6 [66], CD15 [67], CD44 [68]).

Tumorospheres are derived from human GBM tissue (Figure 2). The tissue is first mechanically and/or enzymatically dissociated and filtrated to obtain a single-cell suspension [69,70]. A cell-sorting step can be included to enrich the cell suspension in CSLCs by fluorescence-activated cell sorting or magnetic-activated cell sorting [11,64,65,66,67,68]. Then, cells are suspended in serum-free medium to select only the cells with stem-like cell properties [11,71,72]. The other non-CSLCs of the cell culture are eliminated over the following passages. Tumorospheres form primary spheres and, after a first passage, they grow as secondary spheres. Their culture medium generally contains a mix of glucose, amino acids, inorganic salts, and vitamins. It is usually composed of DMEM alone or in combination with Ham’s F-12 Nutrient Mixture (F12), and supplemented with growth factors, most commonly Epidermal Growth Factor (EGF) and/or basic Fibroblast Growth Factor (bFGF). These factors promote proliferation and maintenance of gene expression characteristics observed in human patient samples [73]. These media can also be supplemented with N2, whose main component is putrescine (a diamine resulting from amino acids decomposition), and B27, which contains a variety of lipid compounds (e.g., linoleic acid, corticosterone, and progesterone). Cell density is a critical parameter to generate tumorospheres [74] and a maximum of 7000 cells/mL should be used to avoid aggregates formation, as these can compromise clonal expansion [11]. However, based on our personal experience, the density can be doubled to enhance tumorospheres generation.

After inducing their differentiation, cells from the tumorospheres can give rise to neural cells (neurons, astrocytes, or oligodendrocytes) identical to the ones found in situ in the parental tumor [11]. Moreover, after orthotopic xenograft, this model recapitulates, at each in vivo passage, the hierarchical cell organization and heterogeneity of the parental tumor [75]. As for the spheroid model, the growth of the tumorospheres is standardized, and a gradient of proliferation, oxygen, nutrients, and pH can be identified [76].

As GBM relapses are known to be associated with GBM CSLCs, tumorospheres are essential to study stemness properties like clonogenicity, proliferation, differentiation, and migration. For example, we used this model to study CSLCs organization and differentiation according to hypoxia and drug response [76]. As CSLCs are resistant to conventional therapies, this model is useful for new drug testing [11,76,77,78].

The tumorosphere model has the advantage to be representative of the cellular organization and the genetic of patients’ tumors, and conserves the molecular subtype of the parental tumor [79]. The genetic stability of the tumorospheres should be checked throughout the culture to control their drift although, according to our experience, they are genetically stable from one passage to another. However, the passage number can affect results because the initial cell populations capable of generating tumorospheres may include transient amplifying cells [74]. In addition, most of the tumorospheres used in the literature are derived from the sorting of a specific cell population, which could add bias. Probably the greater limitation of this model is the lack of cells of the GBM microenvironment, which prevents the study of interactions between CSLCs and other neighboring cells in vivo.

## 3. Organotypic Cultures

### 3.1. Organotypic Slice Model

The organotypic slice model consists in culturing GBM cells or spheroids/tumoroids on a slice of healthy brain to preserve the cerebral cytoarchitecture.

This model, initially set up to study glial tumor cell invasion with conditions mimicking those of the normal brain [80], was inspired by the works of Yamamoto and collaborators based on normal brain slices [81].

To establish this model, healthy brains are needed and mainly mouse and rat brains are used for this purpose (Figure 3). Alternatively, Heiland and collaborators used human healthy brain coming from the periphery of the tumor [82,83]. After extraction of the rodent brain out of the skull, the brain is cut following the coronal axis, into slices of 200 to 400 µm thickness with a vibratome or a tissue chopper. It is possible to embed the brain into low melting agarose [84] or to directly stick it to the platform before cutting [85]. During slicing, it is recommended to bubble a 95% O_2_/5% CO_2_ gas mixture into the vibratome reservoir with cold Phosphate Buffered Saline (PBS), neurobasal medium or hibernate-A medium supplemented with antibiotics/antimycotics [82,83,84,86]. Slices are then transferred into 6-well plates on a cell culture insert [82,83,84]. The slices grow at the interface between the medium and the air. The most commonly used media are composed of neurobasal medium [82,83,86] or EMEM [84] or DMEM [87,88,89] supplemented with penicillin/streptomycin or antibiotic/antimycotic. The addition of horse serum [85,89], F12, HEPES (used to buffer cell culture media), glutamine, or other growth factors as B27 [82,83,84,86] varies from one assay to another and depends also on the type of grafted cells [85,89]. The slice preparation may induce an acute local inflammation that needs to be resolved before grafting. Twenty-four hours after putting the slices in culture, isolated cells or spheroids can be transplanted on the slice. Spheroids can be directly dropped on the top of the slice, or a small incision can be made on the slice to deposit the spheroid into it. Isolated cells can be injected a few microns under the surface of the slice [82], eventually using a microinjection pump [84]. The slices can be kept alive approximately 4 weeks. To facilitate discrimination between slice cells and tumor cells, one cell type should be fluorescent. As an example, we stain the slices and/or transplanted cells with PKH67 or PKH26 tracking dyes. Slice cultures conserve the presence of vessels [87], microglial cells [84,90] and astrocytes [82]. Furthermore, thin structures can also be observed such as microtentacles and filopodia [91].

This model has been highly used to study GBM cells migration and invasion [89]. As the vessels structure is maintained, migration of glioma cells along blood vessels can be investigated [87]. It allows the study of the effect of drugs, genes or proteins present in tumor cells or in the microenvironment, on GBM cells migration and invasion [85,92,93]. Furthermore, the organotypic model enables investigation of features of the immune response such as activation of microglia cells and their role in tumor growth [84]. For example, by using clodronate depleting microglia, Hu and collaborators demonstrated that microglia was necessary to versican stimulation of glioma growth [90]. Using the same approach, Heiland and collaborators showed that the immunosuppressive role of astrocytes in GBM depends on their interaction with microglia [82]. The use of slices from different areas of the brain allows to more precisely evaluate the impact of the brain microenvironment on tumor growth (i.e., cerebellum [85]; subependymal zone [84]; striatum [94]; hippocampus [95]). Cell morphological analyses are also possible to investigate, for example, the correlation between GBM cell morphologies and their invasion rate [94].

Organotypic cultures allow the manipulation of both tumor cells and the brain microenvironment by treating the slice with small molecule inhibitors or by using different genetically engineered mouse models as donors, providing a diversity of applications. By depleting microglia in the slice, this model can be also used to investigate the role of these resident immune cells on tumor growth and drug response. This ex vivo model has been described as a tractable and robust model, less expensive and less time consuming than in vivo models, with a great potential to unravel GBM pathophysiology and drug discovery [96].

By providing ex vivo access to the brain tissue architecture and its complex stroma, this model has the advantage to mimic the interactions between tumor cells and the adjacent non-neoplastic brain microenvironment while still enabling direct observation and cell manipulations in the culture dish [97]. In addition to a complex brain cytoarchitecture, it brings cellular heterogeneity with the presence of local immune cells, blood vessels and neural cells. However, this model lacks specific tumor microenvironment, as only tumoral cells are grafted.

### 3.2. Explants

The explant model consists in culturing small pieces of tumor in plate dishes. Thus, cancer cells and tumoral microenvironment are cultivated together. The explant model was first established in 1929 by R. C. Buckley to characterize the intra-tumoral diversity [6], but was revived in the sixties by several teams.

Accessibility to fresh GBM samples is the main issue to overcome to establish the explant culture. Then, the success of this culture depends on the quality of the tumor pieces received and selected for culture (Figure 4a). Tumor samples must be manipulated within 24 h after resection to avoid tumor degeneration. Necrosis, tissue burnt by the surgery, big vessels and non-neoplastic peri-tumoral brain should be removed before plating the explant [98]. Instead of a single piece of tumor, the residues of surgical vacuuming can be also used to generate explants. In this case, tissue fragments must be very carefully filtered and cleaned with several PBS baths. Then, the tissues are cut into 500 µm^3^ pieces and plated on glass precoated coverslips (12 mm in diameter). Explants have to adhere to the plate before being covered with medium. Several coatings have been used and they have an impact on cellular fate (poly-(L)-lysine [99]; a film of chicken plasma [14]; poly-ornithine [100]; collagen, laminin, gelatin sponge foam or a Millipore filter [15]). Explants are usually cultivated in DMEM or in neurabasal serum-free media supplemented with N2 and B27 [101], or with FBS [15,98,102,103]. To better conserve GBM cellular heterogeneity, our team is currently using a stem-like cell medium supplemented with EGF, bFGF and B27. One or several explants can be plated in the same well, as a sufficient number of fragments is necessary to facilitate the metabolism of the individual fragments [102]. After 72 h of culture, cells start to leave the explant, migrate radially and initiate a “sun shape” formation. Then, the cells continue to proliferate and to migrate, invade the plate, leading sometime to the complete disappearance of the initial core of the explant. Seven to 14 days are needed for a well-established culture (Figure 4b). The medium has to be changed every 72 h and debris must be removed throughout the culture. The explant culture may be maintained for several weeks depending on the medium used, the parameters studied and the growth kinetics of the sample. Based on our experience, the success rate of the explant culture is approximately 50%.

In addition to cancer cells and CSLCs, vessels, fibroblasts, and immune cells have been found in explant cultures [15,76]. Throughout the culture, and depending on the medium used, differentiation can be observed. During the first 2 weeks an increase in GFAP staining and a decrease in the stem-like cell markers as A2B5 can be observed [99,103,104]. Then, in the second period of growth, cells become more differentiated into astrocytes, which co-express GFAP with vimentin, nestin or S-100 and even into oligodendrocytes [99,100]. Some authors described a dedifferentiation phase from the third week of culture and onward, with spindle and epithelioid cells becoming predominant [103]. Cell division by mitoses is commonly seen in the cultures [6,98]. Explant analysis revealed vascular endothelial proliferation but a lack of microvascular proliferation as well as pseudopalisading, probably because areas of necrosis were removed during the first step of the tissue dissection [15]. Hypoxic areas can also be detected within explants by immunostaining of adrenomedullin, a target gene of HIF1α. We observed that hypoxic areas increased when explants were cultivated under hypoxia [76].

The explant model has been initially used to understand patterns of histologic organization and to study the biological properties of GBM cells [15,105]. Radial migration of cells from the explant makes the model particularly useful to study invasion and migration [76,78,99]. Explants have also been used to analyze tumor growth, cellular proliferation, differentiation, stemness and, to a lesser extent, drug sensitivity [76,78,99]. As explants reflect GBM cellular heterogeneity, we used this model to analyze cellular composition and organization upon different microenvironmental conditions and treatments [76,78].

The conservation of the tumor microenvironment and of the tumoral architecture is the main advantage of this model. This allows the study of the impact of the environment on cellular behaviour and microenvironment cell composition. Furthermore, as several pieces of the tumor are cultured in the same well, this model reflects patients’ intra-tumoral heterogeneity. This represented heterogeneity is an advantage for clinical translation, and explants can therefore be a useful tool for personalized medicine. On the other side, experiments using explants must be carefully selected because standardization and reproducibility might be a challenge. The main limitation of this model is probably the lack of the healthy surrounding tissue.

## 4. Organoids

### 4.1. Tumoroids

The tumoroid model aims at reconstituting a tumor in a dish for a long-term culture. This model can be achieved either by the direct expansion of a small piece of tumor or by culturing dissociated CSLCs. The tissue/cells will proliferate and form a small tumor, retaining many key features of their corresponding parental tumor.

The first GBM tumoroid model was initiated by Bjerkvig and collaborators in the 90′s [19] and remained the gold standard [20,25] until the more recent works of Jeremy Rich’s [16] and Hongjun Song’s [17] teams were published.

Tumoroid formation is based on the culture of small pieces of dissected tumors [17,19] or of dissociated cells [16] (Figure 5). Three major culture techniques have been used to produce tumoroids: (i) on an agar coated plate with EMEM supplemented with serum [19,20,24,25]; (ii) into Matrigel in neurobasal serum-free medium supplemented with EGF, bFGF and B27 [16]; (iii) without any matrix in DMEM/neurobasal medium (1:1) supplemented with N2 and B27 [17]. In all protocols, tumoroids are cultivated under shaking. Tumors can be processed into hibernate A medium to preserve the tissue before culturing. Red blood cells must be depleted before culturing dissociated cells by brief hypotonic lysis. Tumoroids can be derived from distinct tumor regions of the parental tumor based on MRI imaging (invasive FLAIR region, contrast enhanced tumor zone, inner necrotic/hypoxic core [16]). The kinetic of growth in vitro depends on the tumoral region and the culturing method used. Tumoroids can be stable and viable in culture for more than a year even if the growth rates slow down after several months [16]. The success rate is between 30% and 90% depending on the technic used [17,25]. Tumoroids can be frozen for later use or for the constitution of a biobank. After defrosting, tumoroids exhibit a growth similar to that of fresh cultured tumoroids and characteristics similar to those of their corresponding parental tumors [17].

These tumoroids preserve the histological characteristics of the primary tumor, harbor similar cellular and nuclear atypia, mitotic rates, fraction of proliferative cells, pleomorphic nuclei, and tissue organization [16,17,19,20,25]. Tumoroids derived from tissue are described to contain oligodendrocyte precursor-like, astrocyte-like, oligodendrocyte-like, and neuron-like cells. Capillaries/CD31^+^ vasculature, fibroblasts, striated collagen fibers, quiescent cells and even immune cells (macrophages/microglia and T cells) can also be observed [17,19,106]. Tumoroids derived from dissociated cells are a mixture of cellular areas composed of CSLCs and non-CSLCs, and non-cellular areas filled with fluid or extracellular matrix. Sex determining region Y-box 2 (SOX2) positive CSLCs have been observed near the core (quiescent cells) but also at the periphery (proliferative cells). A hypoxic gradient was observed within all types of tumoroids [16,17]. Similarities were also found at the transcriptomic and genomic levels between primary tumor and tumoroids. Tumoroids maintain intra- and inter-tumoral molecular heterogeneity and cellular heterogeneity [17].

Not many GBM tumoroid models have been described so far. However, these studies showed that these models offer several advantages such as the possibility to perform molecular investigations. Tejero and collaborators showed that hypoxia and TGFβ signaling promote the proliferation of GBM quiescent cells by inducing a molecular shift from a proneural to a mesenchymal signature [106]. Furthermore, tumoroids derived from distinct tumor regions were set up to evaluate the specific tumorigenic potential of each area after orthotopic xenograft [16]. Tumoroids have also been employed to test responses to standard of care therapy and targeted treatments, including drugs (EGFR, MEK, and mTOR inhibitors) and irradiation [17]. Hubert and collaborators irradiated GBM tumoroids and correlated their radioresistance with the presence of CSLCs [16]. This model has been proposed to test personalized medicine because it can be implemented very quickly, providing results within a time frame compatible with the patient’s short survival expectancy after diagnosis. In line with a personalized therapy, Chimeric Antigen Receptor-T (CAR-T) cell immunotherapy has been tested in a tumoroid model [17]. Therefore, the tumoroid model could be useful to better understand pathology of each individual and help its prognosis [19].

Moreover, the organoid model is the model of choice for long term cultures and to maintain in vitro human tumors.

Tumoroids offer the advantage to grow fast, allowing the establishment of personalized assays and the detection of mixed responses usually discovered only in clinical trials. Nevertheless, they present some limitations. Depending on the culturing method and particularly tumoroids derived from sorted CSLCs, lack of cells of the microenvironment. Moreover, due to their heterogeneity, tumoroids results might not be reproducible and might be not practical for high-throughput screening assays [16].

### 4.2. GBM-Derived from Cerebral Organoids

The purpose of this model is to form or reconstitute GBM organoids in a dish, by following the developmental steps.

This human GBM *3D* culture system was inspired by the pioneer works of Lancaster and collaborators in 2013 and 2014 to generate cerebral organoids in vitro from human embryonic stem cells (hESCs) and induced pluripotent stem cells (hiPSCs) [107,108]. These cerebral organoids reproduce aspects of human cortical development, and several brain regions can be observed.

To produce a GBM organoid, the first step relies on the generation of a cerebral organoid by using the Lancaster’s method. hESCs or hiPSCs cells are cultured in low-bFGF and ROCK (Rho-associated protein kinase) inhibitor medium. Then, embryoid bodies are grown in neural induction medium containing DMEM-F12 supplemented with N2, glutamax and MEM-Non-Essential Amino Acids (MEM-NEAA) [107,108]. After neural induction, an electroporation is performed to introduce single or combined clinically relevant mutations or amplifications of oncogenes/tumor suppressor genes (e.g., *NF1*, *PTEN*, *TP53*, *EGFR*, etc.) by using CRISPR/Cas9 technology. Electroporation can be performed at different times of organoid development depending on the question raised. Modified organoids are then embedded into matrigel and cultivated under shaking into differentiation medium (1:1 mixture of DMEM/neurobasal medium supplemented with N2, B27 without vitamin A, 2-mercaptoethanol, insulin, glutamax and MEM-NEAA) [26,27,28] (Figure 6). These organoids reproduce to some extent the in vivo structural organization of GBMs. They contain both tumor cells with the specific genetic mutations introduced, and normal cells mainly derived from the neural lineage.

This model is a useful tool to explore the developmental natural history of cancer ex vivo [26]. Thanks to this model, the chronology of the mutational steps involved in gliomagenesis can be dissected. This model is also suitable to study effects of target therapy on tumors with specific driver mutations. More generic analyses, such as those of tumor proliferation, invasion, and progression, can be also performed. Furthermore, interactions between tumor cells and non-neoplastic cells can be investigated as they are cultured in the same culture dish.

The presence of non-modified neural cells in this model is an advantage because they serve as safety entities for drug testing. Moreover, this ex vivo tumoral growth allows microscopic observation of tumor development at all stages. This system lacks, however, vascular structures and other cells of the microenvironment, such as microglia [28] which is an important limitation. Therefore, histological characteristics usually found in GBMs such as microvascular proliferation and palisading necrosis cannot be observed.

## 5. Discussion

Clinical studies are time and money consuming. In the case of GBM, many have shown to be sterile or abortive. Their development is based on results obtained on several preclinical animal studies: however, animal models barely reproduce the human physiopathology, and therefore fail to predict the human response. *3D* in vitro models are an alternative to mimic cancers and could shorten this gap. Several *3D* models derived from human tissue are now available to investigate GBM initiation and growth, key driver mutations of gliomagenesis or genes involved in drug sensitivity. Nevertheless, these *3D* models must be used with caution when it comes to clinical relevance. For example, the use of spheroids should be limited as this model is usually generated from non-primary cell lines. Similarly, non-primary cell lines are used to generate tumorosphere-like. These tumorosphere-like should not be used to study stem-like cell properties as they are not proper tumoropheres derived from primary GBM tissues. However, the use of these models can be complementary. For example, for drug testing tumorospheres can firstly be used for a large drug screening, and then explants or organoids can be used for validation of the best hits and for a more personalized medicine. In Table 3, we summarized the recommendations regarding experimental possibilities offered by each model.

Knowing the possibilities and advantages offered by the *3D* culture, the utility of *2D* cell culture can be questioned. The *2D* culture is easy to handle, cost-effective and cells can be expanded rapidly and in big quantities. Since the transcriptomic profile of cells cultured in *2D* or in *3D* shows huge differences, *2D* culture should be used carefully [109].

*3D* models are continuously in development to better resemble GBM tumors. To mimic cellular heterogeneity more closely, co-cultures have been developed. Most of the *3D* models discussed above have been implemented with cells of the microenvironment to generate co-cultures able to overcome the *3D* model’s limitations. Co-cultures of spheroids, tumorospheres, organotypic slices and GBM-derived from cerebral organoids have been used to study the relationships between tumor cells and immune or endothelial cells, neurons, astrocytes, microglia, and macrophages [25,27,32,33,110,111,112,113,114]. Furthermore, thanks to *3D* bioprinting, forms and physical constraints can be applied to biomaterials, which will improve co-cultures experiments. This method combines cells, growth factors, and matrixes to simulate natural tissue features. *3D* bioprinting is useful especially to study cellular interactions, migration, invasion, and drug testing [115,116].

One promising and emerging model is the “tumor-on-chip” model. This model was not detailed in this review because it is not a proper *3D* GBM model but rather a sophisticated co-culture spheroid model [117]. Spheroids can grow in a microfluidic chamber or in a hydrogel matrix, they can be co-cultured with healthy tissue and even vascularized. The chip is made of several reservoirs connected by microchannels. This microfluidic device recapitulates relevant features of tumor physiology and has the advantage of delivering nutrients or therapeutic molecules continuously. This model is highly promising for high-throughput drug screening, prolonged drug release, and mimicry of the blood-brain-barrier, and can be combined with *3D* bioprinting to increase experimental possibilities [118,119].

Another dimension can be added to these *3D* cell culture models: time. Recent advances in time-lapse microscopy, where cells are recorded over time, have started to gain ground. In combination with *3D* models, this leads to *4D* culture models, providing unparalleled new insights.

## 6. Conclusions

All 3D models described in this review offer excellent tools to better understand GBM biology and predict drug response. However, there are still limitations to fully meet all the in vivo GBM criteria and reduce the gap that leads to clinical trials failures. These models propose a reproduction either of tumor cytoarchitecture and cellular heterogeneity or the surrounding healthy brain microenvironment. The combination of both can open the possibility to create and elaborate more accurate models.

## Figures and Tables

**Figure 1 cancers-12-02347-f001:**
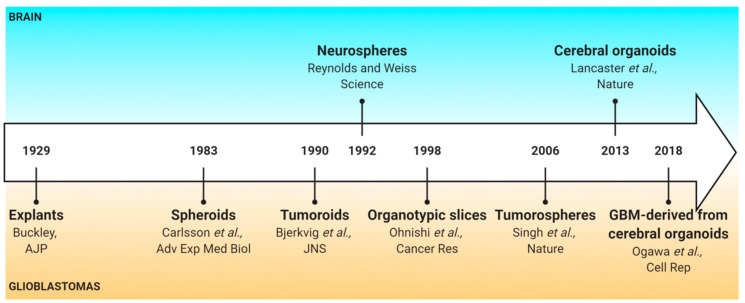
History of pioneer papers on *3D* human glioblastoma models.

**Figure 2 cancers-12-02347-f002:**
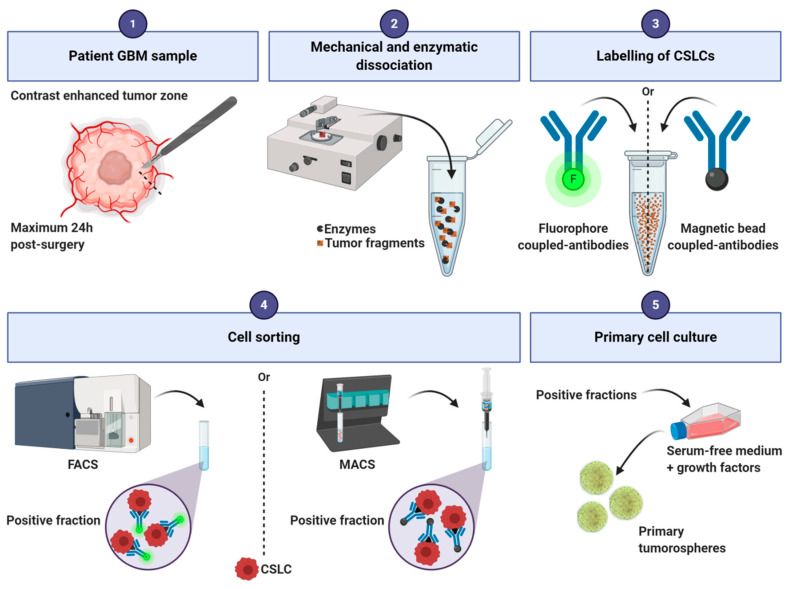
Main steps of the sorting protocol for human glioblastoma tumorospheres production. CSLCs: cancer stem-like cells; FACS: Fluorescence-activated cell sorting; MACS: Magnetic-activated cell sorting.

**Figure 3 cancers-12-02347-f003:**
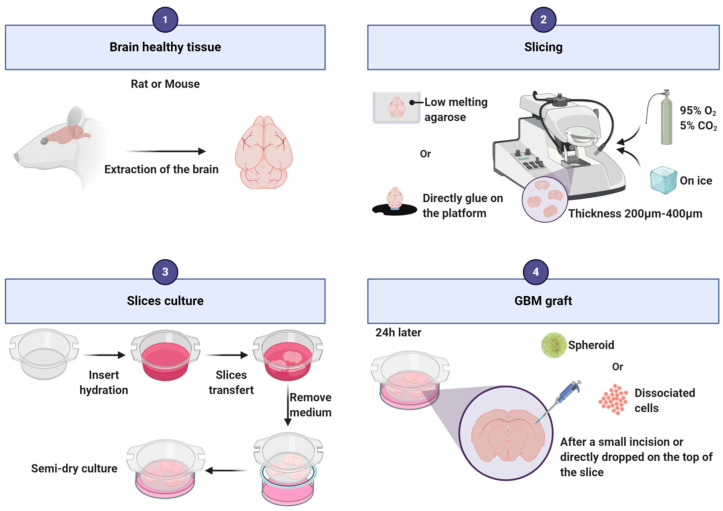
Main steps of the protocol for glioblastoma organotypic slice generation.

**Figure 4 cancers-12-02347-f004:**
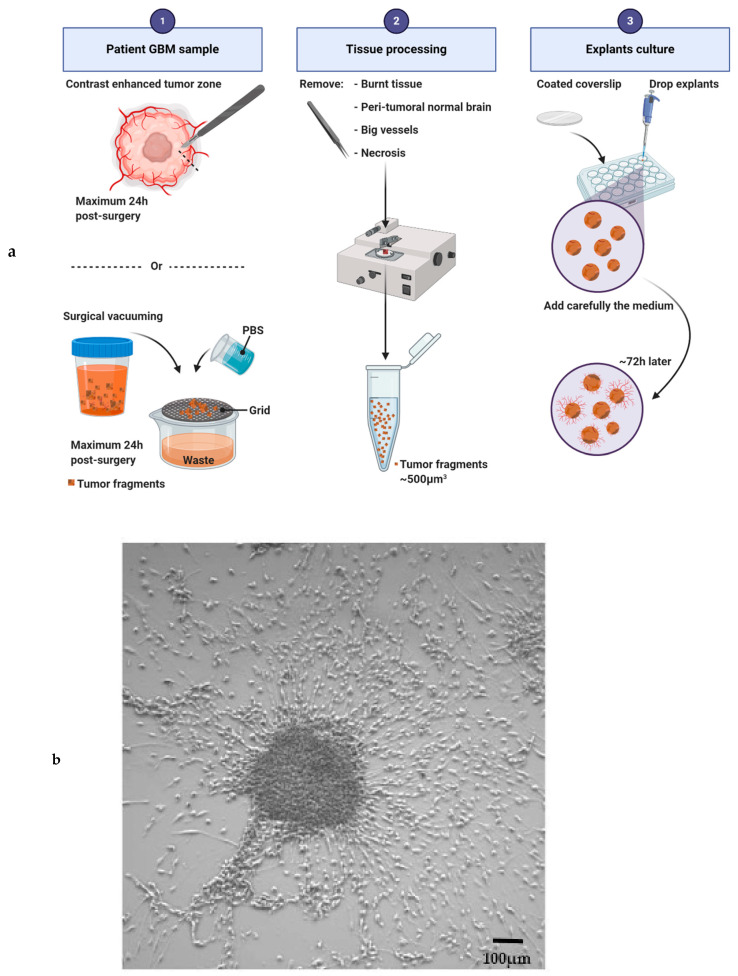
Explant culture and sun shape formation. (**a**) Main steps of the protocol for human glioblastoma explant generation. (**b**) Explant with a sun shape after one week of culture. Scale bar: 100 μm.

**Figure 5 cancers-12-02347-f005:**
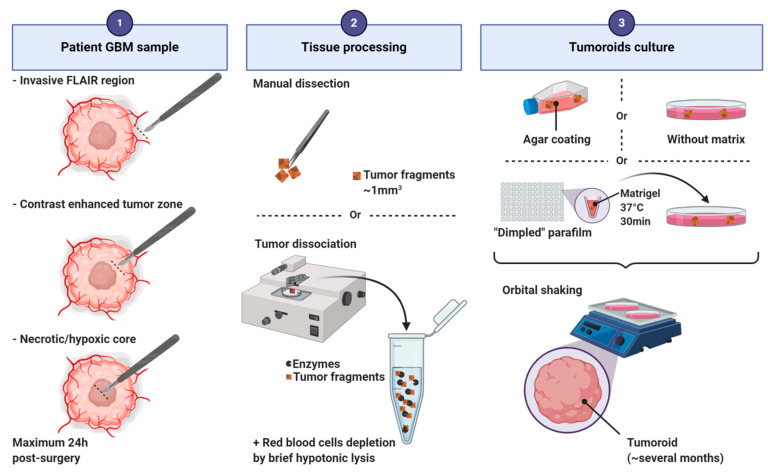
Main steps of the protocol for human glioblastoma tumoroid generation.

**Figure 6 cancers-12-02347-f006:**
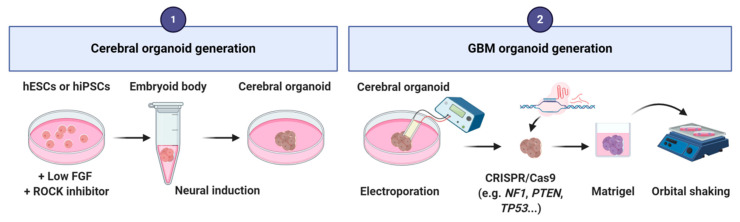
Main steps of the protocol for glioblastoma-derived from cerebral organoids generation.

**Table 1 cancers-12-02347-t001:** *3D* human glioblastoma models terminology found in the literature.

*3D* Models	Material	Alternative Names
Spheroids	Established glioblastoma cell lines	Spheroids [8] Neurospheroids [9] Multicellular tumor spheroids [7] Multicellular aggregates [10]
Tumorospheres	Primary glioblastoma stem-like cells	Spheres [11] Oncospheres [12] Neurosphere-like [13]
Organotypic slices	Grafted isolated cells or spheroids/tumorospheres on brain slices	-
Explants	Glioblastoma tissue	Tissue culture / Tissue particles [14] Organ culture [15]
Tumoroids	Glioblastoma tissue	Organoids [16] Glioblastoma organoids (GBO) [17] Spheroids [18] Multicellular tumor spheroids [19] Biopsy spheroids [20] Fragments spheroids [21] Primary spheroids [22] Patient-derived spheroids [23] Organotypic spheroids [24] Organotypic multicellular spheroids [25]
GBM-derived from cerebral organoids	Embryonic stem cells and induced pluripotent stem cells	Organoids [26] Organoid glioma (GLICO) [27] Neoplastic cerebral organoids (NeoCOR) [28]

**Table 2 cancers-12-02347-t002:** Main methods used to generate spheroids.

Method	Procedure	Matrix	Advantages	Disadvantages	Ref.
Liquid overlay	Tumor cells are placed on tissue culture plastic covered with a thin layer of inert substrate.	Agar Agarose PolyHEMA	Easy-to-use protocol; Easily promotes the aggregation of cells to become spheroids; Co-culture ability; High reproducibility; Inexpensive; Easy to image.	Difficulty to monitor the number and size of spheroids; Heterogeneity of the cell lineage; Lack of interactions between cells and matrix.	[7,34,47]
Ultra–low attachment plates	Cells are seeded in an ultra-low attachment plate without coating as the polystyrene surface offers low adhesion properties.	-	Capability to produce one spheroid per well; Spheroids have a more compact structure than those on agar-coated plates; Easy to image.	Difficulty to monitor the number and size of the spheroids; Heterogeneity of the cell lineage; Lack of interactions between cells and matrix	[48]
Hanging drop method	Cells are dropped in a small volume in the petri dish lid. The lid is subsequently inverted, and aliquots of cell suspension turned into hanging drops without dripping due to surface tension.	-	Easy-to-use protocol; Consistent size and shape controlled by adjusting the density of cell seeding; High reproducibility; Inexpensive; Easy to image.	Heterogeneity of cell lineage; Lack of interactions between cells and matrix; Limited volume of the cell suspension; Difficulty in changing the culture medium.	[49]
Hydrogel embedding/Scaffold	Microcapsules with matrix /cells obtained from cells resuspended in hydrogel *3D* structures that are constructed from a wide-range of materials and possess different porosities, permeabilities, surface chemistries, and mechanical characteristics.	Alginate Matrigel Methylcellulose Collagen Gelatin Silk Chitosan	Large variety of natural or synthetic materials; Customizable; Co-culture possible; Resemble natural extracellular matrix; Circulation of nutrients and cellular waste in and out of the hydrogels.	Deficiency in gelation kinetic control; Undefined composition in natural gels; May not be transparent; Difficulty to remove cells.	[50,51]
Spinner flask bioreactor	Cells are inserted into a chamber with continuous agitation (by gently stirring, rotating the chamber, or perfusing culture media through a scaffold using a pump system). Bioreactors are equipped with media-flowing systems to provide nutrient circulation, metabolic waste removal, and homogeneity of the physical and chemical factors within the bioreactors.	With or without matrix	Easy-to-use protocol; Great spheroid formation; Precise control system and guaranteed reproducibility; Motion of culture assists in nutrient transport; Large scale production.	No control of the cell number/size of spheroids; Cells possibly exposed to shear force in spinner flasks; Specialized equipment required.	[52]

**Table 3 cancers-12-02347-t003:** Experimental possibilities offered by each *3D* model.

Experimental Possibilities		Sphere-Based Models		Organotypic Models		Organoids	
		Spheroids	Tumorospheres	Organotypic Slices	Explants	Tumoroids	GBM-Derived from Cerebral Organoids
CHARACTERISTICS	Success rate	100%	60%	n.s	50%	30 to 90%	n.s
	Heterogeneity in tumor cells/in the peritumoral microenvironment cells	−/−	−/−	−/+	+/−	+/−	−/−
	Genetic stability	−	+	−	+	+	+
	Cryoconservation	+	+	−	−	+	n.s
	Lifespan of the culture	Indefinitely	Indefinitely (1)	4 weeks	<3 weeks	>1 year	n.s
	Standardization	+	+	−	−	−	−
	Patient specific	−	+	+ (2)	+	+	+ (3)
	CSLCs	−	+	+ (2)	+	+	+
PARAMETERS STUDIED	Tumor growth	+	+	+	+	+	+
	Tumor invasion-migration model/can be used to study migration-invasion	−/+	−/+	+/+	+/+	−/+	+/+
	Stemness properties	−	+	+ (4)	+	+	+ (5)
	Environmental influence of the tumor/of the healthy surrounding tissue	−/−	−/−	−/+	+/−	+/−	−/+
	Drug testing	+	+	+	+	+	+
	Radiotherapy	+	+	+	+	+	n.s
	Mechanisms of drug resistance	+	+	−	−	−	+
	High throughput drug screening	+	+	−	−	−	−
	Personalized medicine	−	+	+ (6)	+	+	−
	Immune response	−	−	+	+	+	−
	Gliomagenesis process	−	−	−	−	−	+

(1) Passaging every two weeks; (2) Depends on cells grafted; (3) Specific of genetic modifications introduced; (4) Depends on the cells grafted; (5) Depends on the transformed cells; (6) If primary human cells used; CSLCs: Cancer stem-like cells. n.s: not specified.

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
