# Peer review of "Practical Review on Preclinical Human 3D Glioblastoma Models: Advances and Challenges for Clinical Translation"

_cancers, 2020, doi:10.3390/cancers12092347_

Round 1

Reviewer 1 Report

Sauberna et al. in this review have reported in a chronological manner the major findings related to the 3D model pointing out the main protocols for developing 3D in vitro models for GBM studies as well as the terminology which to date is still confusing.
Overall, the review is well organized and using appropriate references cover all the aspects of developing a 3D in vitro model highlighting the strength and limitations of every single approach which could help to understand and define new experimental protocols in the neuro-oncology field.
The pictures (cartoons) reported are well designed which at view clearly and easily help the readers to retrieve the technical aspect as well as the protocols in order to reproduce or at least improve it with new finds. The tables summarize the pro and cons of each aspect to consider during the development of a 3D model.

One of the major challenges to developing a relevant/human 3D in vitro model involves also the finding of growth conditions of cells. The authors have reported the different media/growth conditions but to ensure reproducibility and translational potentials as well as “3R principles", good manufacturing practice (GMP), and growth conditions at different levels which may need to chose in order to mimic the natural environment of cancer cells. With this regard, the authors could report the publications of Civita P et al. 2019(Civita P, M Leite D, Pilkington GJ. Pre-Clinical Drug Testing in 2D and 3D Human In Vitro Models of Glioblastoma Incorporating Non-Neoplastic Astrocytes: Tunneling Nano Tubules and Mitochondrial Transfer Modulates Cell Behaviour and Therapeutic Respons. Int J Mol Sci. 2019;20(23):6017. Published 2019 Nov 29. doi:10.3390/ijms20236017) and later Leite et al. 2020 (Leite DM, Zvar Baskovic B, Civita P, Neto C, Gumbleton M, Pilkington GJ. A human co-culture cell model incorporating microglia supports glioblastoma growth and migration and confers resistance to cytotoxic. FASEB J. 2020;34(1):1710-1727. doi:10.1096/fj.201901858RR) where they developed an in vitro 3D hydrogel-based "all Human" model of GBM incorporating Astrocytes (Civita et al) and later on with microglia (Leite et al.) showing the capacity of derived patients GBM cells to growth with normal cells and also respond differently to drug treatment within hydrogel and under Human Serum supplementation and no antibiotics. Although further studies in using Human Serum are needed, recent publications have demonstrated that HS and FBS differently influence the behavior of cells in culture which may have an impact on experimental results, especially in 3D cultures (https://doi.org/10.1016/j.yexcr.2018.02.017). Also using adult HS may alter the metabolic behavior as well as the shape and morphology of cancer cells (Steenbergen, R., Oti, M., ter Horst, R. et al. Establishing normal metabolism and differentiation in hepatocellular carcinoma cells by culturing in adult human serum. Sci Rep 8, 11685 (2018). https://doi.org/10.1038/s41598-018-29763-2); it has been also demonstrate which HS may alter the responses to chemotherapeutic agents in breast cancer cells (Emerman, J.T., Fiedler, E.E., Tolcher, A.W. et al. Effects of defined medium, fetal bovine serum, and human serum on growth and chemosensitivities of human breast cancer cells in primary culture: Inference for in vitro assays. In Vitro Cell Dev Biol 23, 134–140 (1987). https://doi.org/10.1007/BF02623594). It is worth denoting this aspect considering that the heterogeneity of GBM its not only related to the cancer cells itself but also from the cues from the microenvironment. Altogether these findings may help to standardize the procedure to growth cells with medium supplemented with HS which should be the most appropriate nutrient environment for determining the effects of therapeutic agents on cells as it most closely resembles the in vivo situation.

Minor Revision.
Grammatics related to the word "three D" which is appropriate to change in 3D and Italic.

Author Response

1) One of the major challenges to developing a relevant/human 3D in vitro model involves also the finding of growth conditions of cells. The authors have reported the different media/growth conditions but to ensure reproducibility and translational potentials as well as “3R principles", good manufacturing practice (GMP), and growth conditions at different levels which may need to choose in order to mimic the natural environment of cancer cells.

We thank the reviewer for this pertinent comment.

The co-culture of GBM-derived cells with non-tumoral human cells such as astrocytes and microglia by using hyaluronic acid-gelatin based hydrogel as suggested by the reviewer is actually very interesting. However, in the main text of our review we included only the literature concerning monocultures. We discussed about co-cultures during the discussion part of this manuscript to show that these monocultures can be enriched and improved by co-culturing (lines 407-426).

The reviewer highlighted that human serum could be also used; we therefore completed this part by adding a sentence “Human serum have been also used to better mimic the natural microenvironment of cancer cells” (line 95).

We added the suggested articles concerning only glioblastomas in the bibliography of the review (lines 93 and 411).

2) Grammatics related to the word "three D" which is appropriate to change in 3D and Italic.

It has been corrected.

Reviewer 2 Report

The objective of this review article is to provide comprehensive updates on widely used 3D models in glioblastoma research. The authors have provided details on history, development of different 3D glioblastoma models including advantages and limitations of these models. The authors have provided nice overview figures of each 3D model covered in the manuscript however there are few concerns as listed below. The manuscript could not be considered for publication in the current form for following concerns.   Major concerns:   1) The authors are requested to elaborate advantages and limitations of each 3D model covered in the manuscript.   2) Could authors also please discuss the challenges with these 3D models in terms clinical relevance and recommendations on areas to be improvised in 3D models to address outstanding questions.   3) Relapses in glioblastoma tumors is a common occurrence. Could authors please comment/discuss on feasibility/applicability of 3D models to understand the role of dormant population in tumor recurrence?   4) Could authors please provide their perspective on what research questions could be addressed best with respective 3D models (at the end of each section of respective 3D model) discussed in the manuscript?   5) The authors have provided details on development of each 3D model including its advantages and limitations. However, the manuscript do not have cohesive flow. Could authors please consider to include their perspective/leading statements along with review of literature to connect the gaps between each section of a 3D model being discussed in the manuscript.

6) Could authors please carefully review the manuscript and rephrase the sentences to convey the message or authors perspective clearly. Example: The authors were discussing about limitations of spheroids models and has mentioned "Some genomic and gene expression changes have been observed when compared to primary tissues". If the authors intention was to mention less correlation in the genomic and gene expression changes was observed with the use of spheroids vs. primary gliomas; the authors need to state the same. Similarly, there were several instances where the author's perspective was lost in the way the sentences were phrased. Could authors please carefully review the manuscript and correct such statements.
  7) Could authors please consider to rephrase the sentences conveying the key messages of 3D models covered in the manuscript. These sentences do not convey the message clearly. Few examples are listed below for authors reference.
7A) For Spheroids: "It consists in the culture of immortalized GBM cell lines...." on Ln # 2 of page # 3.
7B) For Tumorospheres: "It consists in a clonal expansion as free-floating spheres in a defined medium" on Ln # 117 of page # 4
7C) For Tumorospheres: "Tumorospheres are formed of cells with the same stem properties than the mother stem cell (symmetric division) and of cells more engaged in a pathway of differentiation (asymmetric division)" on Ln # 118 of page # 4.
7D) Similarly please review the sections of all 3D models discussed in the manuscript and rephrase the sentences.
  8) The manuscript is poorly written with incomplete or abrupt sentences. Few examples are listed below for authors reference and the authors are requested to cerefully review the manuscript and convey the messages with clarity.
8A) "To facilitate analysis, slices and/or transplanted cells should be fluorescent."
8B) "Several studies have described the conservation of physiological microenvironment in slices cultures as the presence of vessels [85], microglial cells [82,88] and astrocytes [80]".
8C) "Morphological analyses are also possible as, for example, the search of a correlation between GBM cell morphologies and the invasion rate [92]".
  9) Could authors please carefully review the manuscript and correct the grammatical and syntax errors. The incorrect choice of words or the grammatical errors makes it difficult to follow the authors perspective at times and interrupts the flow of the manuscript as well. The following are only few examples for authors reference.
9A) For Tumorospheres: "The tumorosphere model relays on the capability of cells with stem-like cell properties to self-renew". If the authors intention were to say "The tumorosphere model relies on", please correct the sentence accordingly.
9B) For Tumorospheres: "a maximum of 7000 cells/ml should be respected to avoid aggregates formation...". If the authors intention were to say "a maximum of 7000 cells/ml was recommended to avoid aggregates formation..." please correct the sentence accordingly.
9C) For Explants: "If the contain from surgical vacuuming is used instead of a single piece of tumor...". It is difficult to infer what authors refer by "contain" in this sentence.
9D) "Explants have to attach to the plate before to be covered with medium"
  Minor concerns:   1) Could authors please be specific with details and avoid vague references. Example: Please replace "some matrix made of collagen I" with "matrix made of collagen I" on Ln # 85 of page # 4. 2) Could authors please consider following a consistent format to describe 3D models instead of "3D or Three D".  

Author Response

We thank the reviewer for all these relevant remarks.

1) The authors are requested to elaborate advantages and limitations of each 3D model covered in the manuscript.

To make the “advantages and limitations” paragraph clearer, in each section we rephrased some sentences to make them more explicit and completed/reorganized this paragraph when necessary. In these paragraphs which appear at the end of each section, we highlighted the main particularities of each model which could be the most helpful in the model selection.

Please find below the exact location of these “advantages and limitations” paragraphs:

Advantages and limitations for the spheroid model can be found page 5, lines 119-125; for the tumorosphere model: page 7, lines 177-186; for the organotypic slice model: pages 9/10, lines 243-249; for the explant model: page 12, lines 304-312; for the tumoroid model: page 14, lines 371-376 and for GBM-derived from cerebral organoid model: page 15, lines 408-413.

2) Could authors also please discuss the challenges with these 3D models in terms clinical relevance and recommendations on areas to be improvised in 3D models to address outstanding questions.

 A paragraph in the discussion part has been added page 15 line 422 to 429 to temper the clinical relevance of the 3D models:

“Nevertheless, these 3D models must be used with caution regarding clinical relevance. For example, the use of spheroids should be avoided as this model is usually generated from non-primary cell lines. Similarly, non-primary cell lines are used to generate tumorospheres after stem-like cell sorting making the model useless. Furthermore, the use of these models can be complementary. For drug testing as an example, tumorospheres can be used at first for a large drug screening, and in a second stage, explants or organoids can be used for validation of the best hits and for a more personalized medicine. In Table 3, we summarized the recommendations regarding experimental possibilities offered by each model. “

3) Relapses in glioblastoma tumors is a common occurrence. Could authors please comment/discuss on feasibility/applicability of 3D models to understand the role of dormant population in tumor recurrence?  

The role of dormant population in tumor recurrence is indeed an important question to address in the field of glioblastoma research. In this review, we did not discuss on the applicability of 3D models in understanding the role of dormant cells in glioblastoma recurrence because literature is very poor about studies in this field. However, some interesting results have been obtained for other cancers as summarized in the review by Pradhan and collaborators (Pradhan S et al., 2018, Journal of Biomedical Engineering). Nevertheless, we introduced the notion of dormant cells and their importance in glioblastoma relapse page 7, lines 171-173 and page 13, lines 354-356.

4) Could authors please provide their perspective on what research questions could be addressed best with respective 3D models (at the end of each section of respective 3D model) discussed in the manuscript?  

The perspectives of each model were provided at the end of each section just before the section “advantages and limitations”. For a better visibility and clarity we rephrased some sentences and we completed these paragraphs when needed.

Please find below the exact location of these “perspective” paragraphs:

For the spheroids model: page 5, lines 114-117; For the tumorospheres model: page 7, lines 171-175; For the organotypic slices model: page 9, lines 235-241; For the explants model: page 12, lines 296-302; For the tumoroids model: page 13/14, lines 352-369; for the GBM-derived mini-brain model: page 14/15 , lines 401-406).

Furthermore, in Table 3 (page 16) we summarized the recommendations regarding experimental possibilities offered by each model.

5) The authors have provided details on development of each 3D model including its advantages and limitations. However, the manuscript do not have cohesive flow. Could authors please consider to include their perspective/leading statements along with review of literature to connect the gaps between each section of a 3D model being discussed in the manuscript.

As each model is independent from one to the other, we have added a sentence at the end of the introduction section, page 3 lines 68-71, to clarify the construction of our review:

“The models were classified according to their initial biological material and their purpose, and were organized into the following three groups: the sphere-based models derived only from established or primary tumor cell lines, the organotypic cultures based on tissue culture, and organoids generated to reconstitute a tumor in a dish.”

Moreover, we tried to include some personal data throughout the manuscript, as also asked by reviewer #3, in order to add our own experience with some of these models.

For the tumorospheres model, page 7 lines 172-173 “We used this model to study CSLCs organization and differentiation according to hypoxia and drug response”.

For the organotypic slice section, page 8 lines 214-216: “To facilitate discrimination between cells from the slice and tumor cells, slices and/or transplanted cells should be fluorescent. As an example, we stain slices and/or cells with PKH67 or PKH26 tracking dyes”.

For the explant section, page 10, lines 269-270: “To better conserve GBM cellular heterogeneity, we are now using a stem-like cell medium supplemented with EGF, bFGF and B27.”; page 12 lines 300-302: “As explants reflect GBM cellular heterogeneity, we used this model to analyze cellular composition and organization upon different microenvironmental conditions and treatments [76]. »

 6) Could authors please carefully review the manuscript and rephrase the sentences to convey the message or authors perspective clearly. Could authors please carefully review the manuscript and correct such statements.  

7) Could authors please consider to rephrase the sentences conveying the key messages of 3D models covered in the manuscript. These sentences do not convey the message clearly.

8) The manuscript is poorly written with incomplete or abrupt sentences.

9) Could authors please carefully review the manuscript and correct the grammatical and syntax errors. The incorrect choice of words or the grammatical errors makes it difficult to follow the authors perspective at times and interrupts the flow of the manuscript as well.

We agree with the reviewer regarding the English writing; we apologize for this. We corrected English carefully to make it more understandable and enjoyable to read.

Reviewer 3 Report

Dear Editor,

thank you very much for giving me the opportunity to review this manuscript.

The paper is well written. The title: "practical review…" fits perfectly with the theme and this review provides a guide to explore the potentiality of the humans 3D glioblastoma models.

In my opinion this paper will be important for researchers as well as for clinicians to better understand the traslationals properties of these models.

I have just one comment on discussion:a brief description of the personal experience of the Authors with these models it could be enrich the paper,  as they have done in the paragraph on Tumorspheres (2.2 pag. 26).

Author Response

A brief description of the personal experience of the Authors with these models it could be enrich the paper.

We thank the reviewer for his positive comment. As the reviewer recommended, we have added sentences about our own experience in 3D human glioblastoma models.

For the tumorospheres model, page 7 lines 172-173 “We used this model to study CSLCs organization and differentiation according to hypoxia and drug response”.

For the organotypic slice section, page 8 lines 214-216: “To facilitate discrimination between cells from the slice and tumor cells, slices and/or transplanted cells should be fluorescent. As an example, we stain slices and/or cells with PKH67 or PKH26 tracking dyes”.

For the explant section, page 10, lines 269-270: “To better conserve GBM cellular heterogeneity, we are now using a stem-like cell medium supplemented with EGF, bFGF and B27.”; page 12 lines 300-302: “As explants reflect GBM cellular heterogeneity, we used this model to analyze cellular composition and organization upon different microenvironmental conditions and treatments [76]. »

Round 2

Reviewer 2 Report

The objective of this review article is to provide comprehensive updates on widely used 3D models in glioblastoma research. The authors have provided details on history, development of different 3D glioblastoma models including advantages and limitations of these models. The authors have addressed all the comments and there are couple of minor concerns as listed below.   Minor concerns:   1) The authors perspective on applicability of spheroid and tumorosphere models is understandable however the authors need to consider rephrasing their conclusion/choice of words used in the Ln# 422-425. Example: For tumorospheres the authors have concluded "non-primary cell lines are used to generate tumorospheres after stem-like cell sorting making the model useless" on Ln # 425. However the authors have contradicted with their conclusion by stating the "tumorospheres can be used at first for a large drug screening" on Ln # 426. The fact is currently there is no 3D model with all features that could allow it recapitulate actual tumor features and each 3D model comes with their limitations. The authors have also discussed these limitations across the manuscript. Could authors please consider rephrasing the sentences in Ln # 422 - 425; to avoid the use of words such as "avoided" for spheroid model or "useless" for tumorosphere model as each of these models are helpful to address certain questions better than other models in their own way regardless of their limitations.   2) Please replace "sevral" with "several" on Ln # 120   3) Please proofread the manuscript for syntax errors.

Author Response

1) The authors perspective on applicability of spheroid and tumorosphere models is understandable however the authors need to consider rephrasing their conclusion/choice of words used in the Ln# 422-425.

We thank the reviewer for his/her remark as this sentence could indeed be confusing. We rephrase it to make it clearer lines 416-418:

Similarly, non-primary cell lines are used to generate tumorosphere-like. These tumorosphere-like should not be used to study stem-like cell properties as they are not proper tumoropheres derived from primary GBM tissues.”

2) Please replace "sevral" with "several" on Ln # 120  

It has been modified.

3) Please proofread the manuscript for syntax errors.

The manuscript has been corrected.